# Fibrosis Burden of Missed and Added Populations According to the New Definition of Metabolic Dysfunction-Associated Fatty Liver

**DOI:** 10.3390/jcm10194625

**Published:** 2021-10-08

**Authors:** Huiyul Park, Eileen L. Yoon, Mimi Kim, Jung-Hwan Kim, Seon Cho, Dae Won Jun, Eun-Hee Nah

**Affiliations:** 1Department of Family Medicine, Uijeongbu Eulji Medical Center, Eulji University School of Medicine, Uijeongbusi 11749, Korea; bliss153@hanmail.net (H.P.); 12thrib@hanmail.net (J.-H.K.); 2Department of Internal Medicine, Hanyang University College of Medicine, Seoul 04763, Korea; mseileen80@gmail.com; 3Department of Radiology, Hanyang University College of Medicine, Seoul 04763, Korea; bluefish010@naver.com; 4Department of Laboratory Medicine, Health Promotion Research Institute, Seoul 07572, Korea; dduddi3755@hanmail.net

**Keywords:** metabolic dysfunction-associated fatty liver, magnetic resonance elastography, non-alcoholic fatty liver disease, significant fibrosis, advanced fibrosis

## Abstract

Recently, the classification of fatty liver and the definition for non-alcoholic fatty liver disease (NAFLD) have been challenged. Herein, we aim to evaluate the burden of hepatic fibrosis in the missed and added populations following the proposal of the new definition of metabolic dysfunction-associated fatty liver (MAFLD) in a health check-up cohort. A total of 6775 subjects underwent both magnetic resonance elastography (MRE) and an abdominal ultrasound at 13 nationwide health check-up centers in Korea. Significant and advanced hepatic fibrosis was defined as ≥3.0 kPa and ≥3.6 kPa in the MRE test, respectively. The prevalence of sonographic fatty liver (FL) was 47.4%. Among the subjects with sonographic FL, 77.3% and 94% are compatible with NAFLD and with the new MAFLD definitions, respectively. Moreover, 72% of FL cases belong to both the NAFLD and MAFLD definitions, whereas 1.4% is compatible with neither. The population compatible with the MAFLD definition has the following coexisting liver diseases: alcohol-related (71.9%), hepatitis B (23.9%), hepatitis C (0.4%), and both alcohol and viral hepatitis (2.8%). The prevalence of significant and advanced hepatic fibrosis is considerable in the MAFLD-only group. However, the prevalence of significant and advanced hepatic fibrosis is similar in the NAFLD-only group, and neither the NAFLD nor MAFLD group compared to healthy controls. The added population (MAFLD-only group), according to the new MAFLD definition, has a higher metabolic and fibrosis burden when compared to those in the missed population (NAFLD-only group).

## 1. Introduction

Traditionally, fatty liver (FL) has been divided into alcoholic FL associated with a significant alcohol intake and non-alcoholic fatty liver disease (NAFLD) that occurs without a significant alcohol intake. Recently, metabolic dysfunction-associated fatty liver disease (MAFLD) has been proposed as an alternative term for NAFLD [1]. MAFLD emphasizes the metabolic risk as an underlying pathophysiology, as reflected in its definition. Additionally, the MAFLD definition is not exclusive to subgroups of patients who share both metabolic risk and non-NAFLD etiologies. Therefore, it is challenging to classify FL according to the newly proposed MAFLD definition, which requires consideration of both metabolic risk and other etiologies previously not included in the definition of NAFLD.

The shift of the definition from NAFLD to MAFLD has not reached a consensus, and its impact should be evaluated in real-life practice. In particular, its impact on the transition of hepatic fibrosis according to the change in the definition has not been fully assessed. Data estimating fibrosis using an appropriate method in a large cohort are still limited. To date, six cross-sectional studies have compared the fibrosis burden between NAFLD and MAFLD in community-based cohorts [2,3,4,5,6,7]. However, little is known about the prevalence and characteristics of populations compatible with the previous definition of NAFLD, but are now excluded from the new MAFLD definition (missed population). Additionally, there is limited information regarding those who were previously not considered to have NAFLD but have been newly included in the MAFLD definition (added population). Understanding the hepatic fibrosis burden in the added and missed populations in MAFLD is important. However, most previous studies based on the general population used predictive models for hepatic fibrosis, such as the Fibrosis-4 or NAFLD fibrosis score, rather than actual fibrosis measurements [2,3,6]. Hence, they may have overestimated the fibrosis burden, especially in MAFLD with an alcohol-related etiology. Moreover, the number of individuals in the NAFLD-only or MAFLD-only groups was not sufficiently large to adequately compare their characteristics, even in studies that used transient elastography for the fibrosis measurement [4,5].

Therefore, in this study, we evaluate the hepatic fibrosis burden of missed and added populations based on the proposal of the new MAFLD definition in a health check-up cohort.

## 2. Materials and Methods

### 2.1. Study Design

This cross-sectional, retrospective study consecutively included subjects who underwent either voluntary or obligatory health check-ups, including magnetic resonance elastography (MRE), at 13 health promotion centers in Korea between January 2017 and May 2020. Medical records of included subjects were also reviewed. This study was approved by the institutional review board of Hanyang University Hospital (IRB no. HY-2021-04-001-001). The requirement for informed consent was waived because of the retrospective nature of the study, and the analysis used anonymous clinical data.

### 2.2. Rationale for Abdominal Sonography and MRE as Health Check-Up

Abdominal sonography is among the basic examinations most widely performed during health check-ups in Korea. It can be performed either based on examinee preference or during obligatory medical examinations provided every 1 or 2 years by certain groups or companies under the Act of Employment. In contrast, MRE is not included in the routine health check-up program. Nevertheless, there are various types of specialized health check-up programs, including MRE, that persons who might want to obtain a more intense check-up may obtain a test with their own decision. In Korea, patients with chronic liver disease are managed using a separate program. The NHIS provides an abdominal ultrasound and alpha-fetoprotein (AFP) test free of charge twice a year as a separate program for patients with chronic liver disease (viral hepatitis B and C, and cirrhosis) [8]. Hence, patients with known chronic liver disease rarely choose MRE at their own expense.

### 2.3. Inclusion and Exclusion Criteria

A total of 8545 people underwent a health check-up between 1 January 2017 and 30 May 2020. Among them, subjects (N = 6775) examined using both MRE and abdominal ultrasonography were included in the community-based MRE and abdominal ultrasound cohort in this study (Figure 1). Subjects who did not undergo MRE and abdominal ultrasonography at the same time or who had missing values in the biochemical tests, which are prerequisites for the assessment of metabolic syndrome except for serum basal insulin level, were excluded.

### 2.4. Clinical Parameters of the Subjects

Routine questionnaires were administered to every examinee during the health appointment. They included self-reported personal medical history, subjective signs and symptoms, and lifestyle information. Information regarding alcohol consumption (i.e., regularity of alcohol intake, number of intakes during a week or a month, and number of bottles for each intake) and history of metabolic risks (i.e., diagnosis and medication for hypertension, diabetes, and dyslipidemia) were also elicited. Anthropometric measurements included waist circumference, blood pressure, height, weight, total fat mass, and lean mass. The weight-adjusted lean body mass was calculated by dividing the total lean body mass by body weight in percent (total lean mass/body weight ×100%). Additionally, fasting serum glucose, total cholesterol, low-density lipoprotein cholesterol, high-density lipoprotein cholesterol, triglyceride, aspartate aminotransferase (AST), alanine aminotransferase (ALT), and γ-glutamyl transferase (GGT) were measured.

### 2.5. Definition of NAFLD and MAFLD

Liver NAFLD was defined as the presence of FL on abdominal ultrasonography without any other chronic liver disease etiologies, such as alcohol-related liver disease, chronic hepatitis B, chronic hepatitis C, and/or autoimmune liver diseases [9]. MAFLD was defined as having one or more of the following criteria: (1) overweight or obesity (body mass index (BMI) ≥ 23 or higher), (2) lean (BMI < 23) but having two or more metabolic risk abnormalities, and (3) type 2 diabetes mellitus (DM) [1]. Metabolic risk abnormalities were defined as follows [10]: (1) waist circumference ≥80 cm for females and ≥90 cm for males, (2) blood pressure ≥130/85 mmHg or higher and/or taking hypertension medication, (3) serum triglyceride ≥150 mg/dL, (4) high-density lipoprotein cholesterol <50 mg/dL for females and <40 mg/dL for males, and (5) fasting glucose level of 100–125 mg/dL and/or taking diabetes medication.

Two groups of subjects were generated with a shift of definition from NAFLD to MAFLD. Of the two, the NAFLD-only group (or missed group) was defined as a group of subjects who were missed from the MAFLD group. The MAFLD-only group (or added group) was defined as a group of subjects who were newly included in the MAFLD group.

### 2.6. Assessment of Hepatic Fibrosis Severity

Liver stiffness was measured using MRE. All MRE examinations were performed on MRE hardware (GE Healthcare) with a 1.5-T imaging system using a two-dimensional MRE protocol [11]. The cut-offs for fibrosis severity were set at MRE values of ≥3.0 kPa (≥fibrosis stage 2 (F2), significant fibrosis), ≥3.6 kPa (≥fibrosis stage 3 (F3), advanced fibrosis), and ≥4.7 kPa (fibrosis stage 4, liver cirrhosis) [12].

### 2.7. Statistical Analyses

Continuous and categorical variables were presented as mean ± standard deviation (SD) and numbers (%), respectively. Categorical variables were analyzed using either the chi-square test or Fisher’s exact test, whereas continuous variables were analyzed using the Student’s independent *t*-test. Statistical significance was set at *p* < 0.05. Statistical analyses were performed using SPSS version 26.0 for Windows (SPSS Inc., Chicago, IL, USA).

## 3. Results

### 3.1. Baseline Characteristics of the Entire Population

A total of 6775 subjects was included in the study (Table 1). Males were more prevalent in our cohort (80.6%). The mean age of the participants was 46.8 years. The prevalence of hypertension, type 2 diabetes, and metabolic syndrome was 28%, 8%, and 22.4%, respectively. The proportion of significant and advanced fibrosis was 8.2% and 2.2%, respectively.

### 3.2. Distribution of Fatty Liver According to NAFLD and MAFLD Definitions

In our cohort, the prevalence of sonographic FL was 47.4%. Among the subjects with FL, nearly 72.7% were compatible with both NAFLD and MAFLD definitions (Figure 2). However, only 77% of these subjects were compatible with the NAFLD definition, whereas over 94% of subjects with FL were compatible with the MAFLD definition. The proportion of subjects with FL belonging to the NAFLD-only group and the MAFLD-only group, according to the new MAFLD definition, was 4.6% and 21.3%, respectively. In the entire population, the prevalence of the NAFLD-only group was 2.2%, whereas subjects in the MAFLD-only group constituted 10.1% of the entire population (Figure 1). The MAFLD-only group had the following coexisting liver diseases: alcohol-related (significant alcohol consumption) (71.9%), hepatitis B (23.9%), hepatitis C (0.4%), and both alcohol and viral hepatitis (2.8%) (Figure 2).

Interestingly, 1.4% of subjects with FL met neither the NAFLD nor the MAFLD criteria. Neither the NAFLD nor MAFLD group was metabolically healthy; 80.4% (36/46) of subjects had significant alcohol intake, 17.4% (8/46) also had HBV, and 2.2% (1/46) had both viral and significant alcohol intake (data not shown).

### 3.3. Clinical Characteristics of the MAFLD-Only Group (Added Population)

The proportion of the MAFLD-only group among the subjects with FL was considerable (21.3%). Except for the amount of alcohol consumption, there were no significant differences in BMI, waist circumference, and the weight-adjusted lean mass between the NAFLD and MAFLD group and the MAFLD-only group. The proportion of significant fibrosis (≥F2) was considerably higher in the MAFLD-only group than in the NAFLD and MAFLD group (13.1% vs. 9.0%, *p* = 0.001), and the NAFLD-only group (missed population) (13.1% vs. 6.1%, *p* = 0.016) (Figure 3A). However, the prevalence of advanced fibrosis (≥F3) did not differ among the NAFLD and MAFLD group, NAFLD-only group, and MAFLD-only group (Figure 3B). The total fat mass was higher in the MAFLD-only group than that in the NAFLD-only group (21.6 ± 5.7 kg vs. 13.6 ± 2.7 kg, *p* < 0.001). In contrast, the weight-adjusted lean body mass in the MAFLD-only group was significantly lower than that in the NAFLD-only group (67% vs. 72.5%, *p* < 0.001) and the control group (67% vs. 70.1%, *p* < 0.001).

### 3.4. Clinical Characteristics of the NAFLD-Only Group (Missed Population)

The proportion of the NAFLD-only group (missed population) among the subjects with FL was only 4.6%. There were no significant differences in BMI, waist circumference, and the weight-adjusted lean mass between the NAFLD-only group and neither the NAFLD nor MAFLD groups. The fibrotic burden was also similar between the two groups. Moreover, the prevalence of significant fibrosis and advanced fibrosis was similar between the NAFLD-only group and the healthy control group. Additionally, the NAFLD-only group showed better profiles of anthropometric data, including a lower BMI and total fat mass, than the control group. The total fat mass was also lower in the NAFLD-only group than the healthy control group (13.6 ± 2.7 kg vs. 16.2 ± 4.7 kg, *p* < 0.001). In contrast, the weight-adjusted lean mass in the NAFLD-only group was significantly higher than that in the control group (72.5% vs. 70.1%, *p* < 0.001).

### 3.5. Sensitivity Analysis According to Various MRE Cut-Offs

The prevalence of hepatic fibrosis at various cut-off values was compared (Table 2), following the sensitivity analysis for the other MRE cut-offs. When ≥3.4 kPa was used as the cut-off for significant fibrosis (≥F2) [13], the groups compatible with the MAFLD definition, regardless of the NAFLD definition, showed a significantly higher proportion of significant fibrosis compared to the healthy controls. Moreover, the MAFLD-only group showed a higher prevalence of significant fibrosis when compared with the control group (4.7% vs. 1.9%, *p* < 0.001). In contrast, the prevalence of significant fibrosis was not different between the NAFLD-only group and the healthy controls. When ≥3.8 kPa was used as the cut-off for advanced fibrosis (≥F3) [13], the proportion of subjects with advanced fibrosis was significantly higher in both the NAFLD and MAFLD groups (1.9% vs. 0.9%, *p* = 0.003) (data not shown) and the MAFLD-only group (1.9% vs. 0.9%, *p* = 0.029) compared to the healthy control group. However, the prevalence of advanced fibrosis between the NAFLD-only group and healthy controls was comparable. The MAFLD-only group alone showed a significantly higher fibrotic burden compared to the healthy control group, regardless of the MRE cut-off.

## 4. Discussion

This is the first study to reclassify patients with sonographic FL following the proposed MAFLD definition. Additionally, we evaluated the anthropometric and fibrotic characteristics of the populations that have been newly added to and have been missed in the MAFLD population using MRE in a large health check-up cohort. Our data were gathered from 13 nationwide health check-up centers. The prevalence of metabolic diseases, such as hypertension, diabetes, and metabolic syndrome, was also consistent with that of the general Korean population [14,15,16].

It is too early to discuss the change of definition regarding NAFLD to MAFLD, and it has not been endorsed by Hepatology Societies. However, it would be valuable to evaluate the impact in real-life practice. In this regard, it is interesting to note that approximately 94% of subjects with sonographic FL satisfied the MAFLD definition. Moreover, we observed a grey area, wherein patients with sonographic FL satisfied neither the NAFLD nor the MAFLD definition.

The fibrosis burden was directly related to the risk of decompensated cirrhosis and hepatocellular carcinoma. Therefore, it would be meaningful to evaluate the impact of this transition on the definition of comorbidities, anthropometric profiles, and fibrosis severity in a community-based population.

The missed population (NAFLD-only group) constituted less than 3% of the studied cohort and showed a similar rate of significant fibrosis as that of the healthy control group. In addition to the fibrosis burden, the body composition of the missed population also showed favorable data compared to the healthy control group. The total fat mass was lower and lean body mass was higher in the missed population than in the control group. The NAFLD-only group showed the most benign features in terms of metabolic risks and anthropometric data when their characteristics were compared between the control and MAFLD-only groups. However, Lee et al. suggested that the risk of cardiovascular events was somewhat higher in this missed population than in the control group [17]. Long-term follow-up data of cardiovascular events in this seemingly benign subgroup should therefore be further evaluated in a community-based population.

The newly added population based on the MAFLD definition (MAFLD-only group) is worthy of medical attention. One-tenth of the population was additionally included in the MAFLD definition, which did not satisfy the conventional NAFLD definition. Moreover, the MAFLD-only group was distinct from the conventional NAFLD population (higher proportion of significant fibrosis). The total fat mass was higher and the lean body mass was lower in the added population than in the missed population, according to the new definition of MAFLD. In addition, the MAFLD-only group showed a clear difference from the NAFLD-only group in terms of the prevalence of comorbidities, fat mass, weight-adjusted lean body mass, and fibrosis features considered in this study, except for age.

This study had several limitations. First, males were more prevalent in the cohort. Moreover, subjects concerned with liver health were more likely to be included in the study, because MRE was offered as an additional option to be tested with their own expense. Even though our health check-up cohort reflected the general population, there is a possibility of selection bias. Nevertheless, the overall prevalence of hypertension (28.0%), DM (8.0%), and metabolic syndrome (22.4%) was comparable with that of the general population. Additionally, there was a scarce possibility that people with chronic liver disease underwent MRE though the health check-up, because almost all of them were managed under a separate program in Korea. Moreover, we assumed that the large number of females (*n* = 1315) and representative characteristics of our nationwide data were sufficient for ensuring an adequate statistical power. However, the effect of sex on FL and metabolic risks should be considered in future studies. Second, there is no consensus on the MRE cut-off values for fibrosis in patients with FL. In this study, the cut-off value was set at 3.0 kPa and 3.6 kPa for significant and advanced fibrosis, respectively. In the sensitivity analysis, the difference in fibrotic burden between the MAFLD-only group and the NAFLD-only group and the NAFLD groups disappeared when ≥3.4 kPa was used as the cut-off value. At the cut-off value of 3.4 kPa, the prevalence of hepatic fibrosis in all subjects decreased to 3.2%. The number of patients was too small for an appropriate comparison between groups when the cut-off value was ≥3.4 kPa. This was also related to the fact that our study was based on the results of the general population undergoing health check-ups. However, the fibrotic burden in the MAFLD-only group was higher than in the control group, regardless of the MRE cut-off. Therefore, we believe that the fibrotic burden in the MAFLD-only group was higher than in any other group.

In conclusion, 94% of sonographic FL cases were compatible with the newly proposed MAFLD definitions. The fibrosis burden of the NAFLD-only group (missed population) was low and was similar to that of the healthy control group. An additional one-tenth of the population with a higher metabolic and fibrosis burden was also compatible with the MAFLD definition.

## Figures and Tables

**Figure 1 jcm-10-04625-f001:**
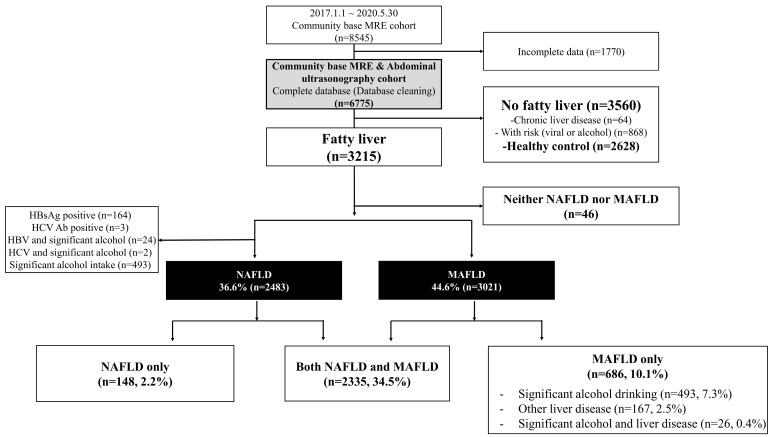
Flow diagram of study. Abbreviations: MAFLD, metabolic-associated fatty liver disease; MRE, magnetic resonance elastography; NAFLD, non-alcoholic fatty liver disease.

**Figure 2 jcm-10-04625-f002:**
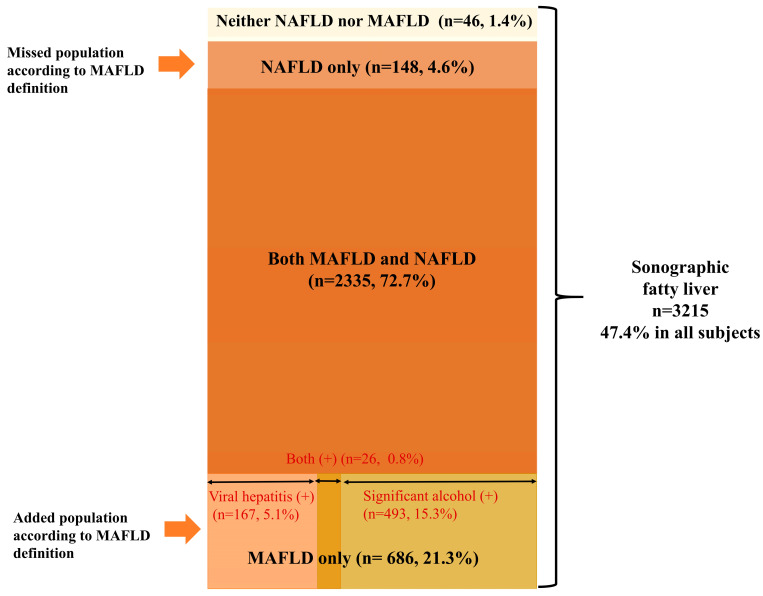
Venn diagram representing the prevalence of fatty liver disease in the community-based cohort, according to the definition of NAFLD and MAFLD. Abbreviations: MAFLD, metabolic-associated fatty liver disease; NAFLD, non-alcoholic fatty liver disease.

**Figure 3 jcm-10-04625-f003:**
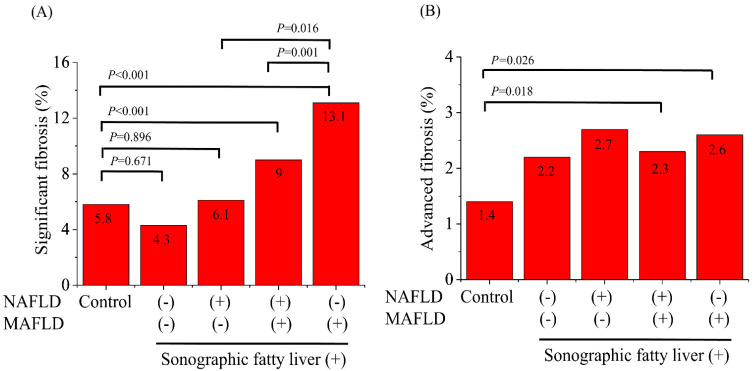
The prevalence of significant fibrosis (≥3.0 kPa) (**A**) and advanced fibrosis (≥3.6 kPa) (**B**) on magnetic resonance elastography among the healthy controls, and those with neither NAFLD nor MAFLD, NAFLD only, both NAFLD and MAFLD, and MAFLD only. Abbreviations: MAFLD, metabolic-associated fatty liver disease; NAFLD, non-alcoholic fatty liver disease.

**Table 1 jcm-10-04625-t001:** Baseline characteristics of patients who underwent MRE and abdominal ultrasound during health check-ups, according to the definitions of NAFLD and MAFLD.

Characteristics	Total (*n* = 6775)	Control (*n* = 2628)	Neither NAFLD nor MAFLD (*n* = 46)	NAFLD (*n* = 2483)	NAFLD Only (*n* = 148)	MAFLD Only (*n* = 686)	Both NAFLD and MAFLD (*n* = 2335)	*p* ‡	*p* §	*p* ‖
Age (years) †	46.8 ± 10.3	46.4 ± 10.9	46.4 ± 10.4	47.4 ± 9.8	46.5 ± 10.2	46.9 ± 9.5	47.5 ± 9.8	0.875	0.228	0.636
Male	5460 (80.6)	1941 (73.9)	39 (84.8)	2229 (89.8)	116 (78.4)	622 (90.7)	2113 (90.5)	0.222	0.487	<0.001
Hypertension	1895 (28)	559 (21.3)	4 (8.7)	916 (36.9)	11 (7.4)	236 (34.4)	905 (38.8)	<0.001	0.230	<0.001
Type 2 diabetes	545 (8)	103 (3.9)	0 (0)	312 (12.9)	1 (0.7)	81 (11.8)	320 (13.7)	0.043	0.435	<0.001
Alcohol consumption (g/week) †	94 ± 166	53 ± 1	335 ± 229	34 ± 54	37 ± 57	304 ± 248	34 ± 54	0.391	<0.001	<0.001
Number of metabolic risks †	1.4 ± 1.3	0.9 ± 1	0.3 ± 0.4	2.1 ± 1.3	0.3 ± 0.5	2.2 ± 1.2	2 ± 1.2	<0.001	0.654	<0.001
Metabolic syndrome	1517 (22.4)	243 (9.2)	0 (0)	946 (38.1)	0 (0)	250 (36.5)	946 (40.5)	<0.001	0.427	<0.001
BMI (kg/m^2^) †	24.8 ± 3.2	23.4 ± 2.6	21.8 ± 0.8	26.4 ± 3	21.6 ± 1.1	26.6 ± 2.8	26.7 ± 2.8	<0.001	0.076	<0.001
Waist circumference (cm) †	85.3 ± 9.1	81.2 ± 8.1	79.3 ± 5.3	90 ± 7.6	78.9 ± 5.1	90.6 ± 7.3	90.7 ± 7.2	0.001	0.085	<0.001
Total fat mass (kg) †	18.5 ± 5.8	16.2 ± 4.8	13.6 ± 2.6	21 ± 5.7	13.6 ± 2.7	21.6 ± 5.7	21.5 ± 5.5	<0.001	0.020	<0.001
Lean mass (kg) †	49.1 ± 8.9	46.9 ± 8.6	44.9 ± 8.2	51.8 ± 8.3	45.2 ± 6.5	52.4 ± 8	52.2 ± 8.2	0.022	0.082	<0.001
Lean mass * 100/BW	68.6	70.1	70.5	67.1	72.5	67	66.8	<0.001	0.563	<0.001
SBP (mmHg) †	116 ± 13	114 ± 13	112. ± 11	119 ± 13	113 ± 10	117 ± 13	119 ± 13	0.234	0.006	<0.001
DBP (mmHg) †	74 ± 9	73 ± 9	71 ± 7	76 ± 9	72 ± 7	75 ± 9	77 ± 9	0.103	0.015	<0.001
AST (IU/L) †	30 ± 19	26 ± 13	27 ± 9	34 ± 22	29 ± 22	34 ± 25	34 ± 22	0.016	0.846	0.022
ALT (IU/L) †	32 ± 33	23 ± 21	27 ± 16	41 ± 35	28 ± 26	42 ± 46	42 ± 36	0.006	0.463	<0.001
GGT (U/L) †	55 ± 84	43 ± 54	43 ± 45	72 ± 112	57 ± 142	62 ± 63	73 ± 110	0.007	0.023	0.559
Triglyceride (mg/dL) †	142 ± 113	112 ± 76	102 ± 52	181 ± 129	105 ± 55	180 ± 160	186 ± 131	0.314	0.815	<0.001
HDL (mg/dL) †	53 ± 12	59 ± 13	55 ± 11	48 ± 10	56 ± 12	48 ± 10	48 ± 10	0.772	0.465	<0.001
Glucose (mg/dL) †	98 ± 20	94 ± 16	92 ± 7	103 ± 24	92 ± 9	103 ± 24	104 ± 24	0.051	0.657	<0.001
Liver stiffness (kPa) †	2.34 ± 0.56	2.26 ± 0.51	2.36 ± 0.4	2.38 ± 0.54	2.26 ± 0.51	2.43 ± 0.62	2.38 ± 0.54	0.99	0.014	0.002
Significant fibrosis	554 (8.2)	153 (5.8)	2 (4.3)	219 (8.8)	9 (6.1)	90 (13.1)	210 (9)	0.896	0.001	0.016
Advanced fibrosis	151 (2.2)	37 (1.4)	1 (2.2)	58 (2.3)	4 (2.7)	18 (2.6)	54 (2.3)	0.204	0.663	0.957

Data are expressed as number (percent). * means Lean mass 100/BW. † Data are shown as mean ± standard deviation. When calculating the *p* value, the *t*-test and chi-square test were used for continuous and categorical variables, respectively. ‡ *p*-value when the control group was compared to the NAFLD-only group. § *p*-value when the NAFLD group was compared to the MAFLD-only group. ‖ *p*-value when the NAFLD-only group was compared to the MAFLD-only group. Abbreviations: AST, aspartate transaminase; ALT, alanine transaminase; BMI, body mass index; BW, body weight; DBP, diastolic blood pressure; GGT, γ-glutamyl transferase; HDL, high density lipoprotein; MAFLD, metabolic-associated fatty liver disease; MRE, magnetic resonance elastography; NAFLD, non-alcoholic fatty liver disease; SBP, systolic blood pressure.

**Table 2 jcm-10-04625-t002:** The prevalence of hepatic fibrosis at various cut-off values.

Cut-Off Values (KPa)	Total (*n* = 6775)	Control (*n* = 2628)	NAFLD Group (*n* = 2483)	NAFLD-Only Group (*n* = 148)	MAFLD-Only Group (*n* = 686)	*p* †	*p* ‡	*p* §	*p* ‖
≥3.0	554 (8.2)	153 (5.8)	219 (8.8)	9 (6.1)	90 (13.1)	0.896	0.001	0.016	<0.001
≥3.2	340 (5)	84 (3.2)	136 (5.5)	6 (4.1)	54 (7.9)	0.556	0.019	0.103	<0.001
≥3.4	220 (3.2)	51 (1.9)	91 (3.7)	5 (3.4)	32 (4.7)	0.226	0.230	0.491	<0.001
≥3.6	151 (2.2)	37 (1.4)	58 (2.3)	4 (2.7)	18 (2.6)	0.204	0.663	0.957	0.026
≥3.8	111 (1.6)	24 (0.9)	47 (1.9)	3 (2)	13 (1.9)	0.179	0.997	0.915	0.029
≥4.2	72 (1.1)	17 (0.6)	29 (1.2)	2 (1.4)	10 (1.5)	0.312	0.542	0.921	0.035
≥4.6	46 (0.7)	7 (0.3)	16 (0.6)	0 (0)	8 (1.2)	0.53	0.163	0.187	0.002

Data are expressed as number (percent). When calculating the *p* value, the chi-square test was used. † *p*-value when the control group was compared to the NAFLD-only group. ‡ *p*-value when the NAFLD group was compared to the MAFLD-only group. § *p*-value when the NAFLD-only group was compared to the MAFLD-only group. ‖ *p*-value when the control group was compared to the MAFLD-only group. Abbreviations: MAFLD, metabolic-associated fatty liver disease, NAFLD, non-alcoholic fatty liver disease; SBP, systolic blood pressure.

## Data Availability

Data available on request due to privacy/ethical restrictions.

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
