# Peer review of "Fibrosis Burden of Missed and Added Populations According to the New Definition of Metabolic Dysfunction-Associated Fatty Liver"

_jcm, 2021, doi:10.3390/jcm10194625_

Round 1

Reviewer 1 Report

The authors have undertaken an important study to show the differences between NAFLD and MAFLD.

This is of importance in view of the increasing use of the new nomenclature. It is relieving to note that for the majority of patients there are no changes.

The design is robust and it is well carried out.

I have only one issue of concern and that is the fact that for many participants the data was obtained from compulsory health checks. I. Am not familiar with this concept but if the participation of the patients was compulsory then there are ethical problems connected with using the data.

This point needs to be addressed by the authors.

Author Response

Reviewer 1

The authors have undertaken an important study to show the differences between NAFLD and MAFLD.

This is of importance in view of the increasing use of the new nomenclature. It is relieving to note that for the majority of patients there are no changes.

The design is robust and it is well carried out.

I have only one issue of concern and that is the fact that for many participants the data was obtained from compulsory health checks. I. Am not familiar with this concept but if the participation of the patients was compulsory then there are ethical problems connected with using the data.

This point needs to be addressed by the authors.

☞ Author responses: Thank you very much for your time and precious comments. We fully understand your concern that MRE is not an essential element for health check-up program. And there may be a possibility of performing compulsory MRE test in purpose of a research.

However, there are various types of specialized health check-up programs that are currently run by major private health promotion centers in Korea regardless of the research. And these retrospective data are widely open to researchers for study with the approval of the institutional review boards. MRE is one the elements of more specialized health check-up programs that persons who might want to get more intense work-up may get a check-up with their own decision. However, this issue has been intensely written in the section of “2.2 Rationale for abdominal sonography and MRE as health check-up” on lines 78~80, page 2 and added a sentence as follows, “In contrast, MRE is not included in the routine health check-up program. Nevertheless, there are various types of specialized health check-up programs including MRE that persons who might want to get more intense check-up may get a test with their own decision. It is offered as an additional option with its own associated expense.”

Reviewer 2 Report

The article presents the results of an original research aimed to evaluate the burden of hepatic fibrosis in the missed and added populations following the change in definition of NAFLD to metabolic dysfunction-associated fatty liver (MAFLD) in a health check-up cohort.

The manuscript is well structured, but as the authors point out, the study has several limitations and possible selection bias.

The topic is interesting. A new definition for NAFLD can be useful and "positive criteria" to diagnose the MAFLD are required. There is currently little evidence in literature and this shift of the definition should be evaluated in real-life practice.

Author Response

Reviewer 2

The article presents the results of an original research aimed to evaluate the burden of hepatic fibrosis in the missed and added populations following the change in definition of NAFLD to metabolic dysfunction-associated fatty liver (MAFLD) in a health check-up cohort.

The manuscript is well structured, but as the authors point out, the study has several limitations and possible selection bias.

The topic is interesting. A new definition for NAFLD can be useful and "positive criteria" to diagnose the MAFLD are required. There is currently little evidence in literature and this shift of the definition should be evaluated in real-life practice.

☞ Author responses: Thank you very much for your comment. We fully agree with you on that there are concerns for need of change in NAFLD definition regardless of application in the real-life practice. Additionally, our study is not free of limitations regarding an indispensable selection bias that MRE is optionally tested in persons who volunteered for the exam with their own expense during the health check-up. However, our study has strengths in that 1) our results were based on a large health check-up cohort that reflects more real-world data than results based on a hospital cohort, 2) our study assessed the hepatic fibrotic burden using magnetic resonance elastography, which is the second most reliable and objective method after the liver biopsy. Therefore, we believe that our study may help readers to get an insight regarding the new MAFLD definition.
